# Fighting Antimicrobial Resistance: Development and Implementation of the Ghanaian National Action Plan (2017–2021)

**DOI:** 10.3390/antibiotics11050613

**Published:** 2022-05-03

**Authors:** Wolfgang Hein, Leslie Mawuli Aglanu, MacDonnel Mensah-Sekyere, Anne Harant, Johanna Brinkel, Maike Lamshöft, Eva Lorenz, Daniel Eibach, John Amuasi

**Affiliations:** 1German Institute of Global and Area Studies (GIGA), 20354 Hamburg, Germany; 2Faculty of Business, Economics and Social Sciences, University of Hamburg, 20146 Hamburg, Germany; anne.harant@web.de; 3Kumasi Centre for Collaborative Research in Tropical Medicine (KCCR), PMB UPO, Kumasi, Ghana; aglanu@kccr.de (L.M.A.); amuasi@kccr.de (J.A.); 4University Medical Centre Groningen (UMCG), University of Groningen, 9713 GZ Groningen, The Netherlands; 5Kwame Nkrumah University of Science and Technology (KNUST), PMB UPO, Kumasi, Ghana; macmesek@gmail.com; 6Bernhard Nocht Institute for Tropical Medicine (BNITM), 20359 Hamburg, Germany; brinkel@bni-hamburg.de (J.B.); lamshoeft@bnitm.de (M.L.); eva.lorenz@bnitm.de (E.L.); 7German Centre for Infection Research (DZIF), 20354 Hamburg, Germany; 8Federal Ministry for Economic Cooperation and Development (Germany), 53113 Bonn, Germany; daniel.eibach@bmz.bund.de

**Keywords:** antimicrobial resistance, AMR politics, national action plan, implementation, governance, transparency, mobilizing finance, awareness, monitoring and evaluation, One Health

## Abstract

In recent years, Ghana has been recognised as a leading player in addressing antimicrobial resistance (AMR) in Africa. However, based on our literature review, we could not ascertain whether the core elements of the national action plan (NAP) were implemented in practice. In this paper, we present a qualitative analysis of the development of AMR-related policies in Ghana, including the NAP. We conducted 13 semi-structured expert interviews to obtain at a more thorough understanding of the implementation process for the AMR NAP and to highlight its accomplishments and shortcomings. The results show that AMR policies, as embodied in the NAP, have led to an extended network of cooperation between stakeholders in many political fields. Broadly, limited allocation of financial resources from the government and from international cooperation have been deplored. Furthermore, the opportunity for using the NAP in mainstreaming the response to the threat of AMR has not been seized. To the general public, this remained hidden behind a number of other relevant health topics such as infection prevention, veterinary services and pharmaceutical regulation. As a One Health (OH) challenge, developing countries could integrate AMR NAPs into other health and environmental programmes to improve its implementation in practice.

## 1. Background

A recent systematic analysis in the Lancet [1] provides estimates of deaths associated with bacterial antimicrobial resistance (AMR) and lists western sub-Saharan Africa as the region with the highest all-age death rates, attributable to and associated with bacterial AMR. The review reiterates the extreme importance of an effective political strategy to fight AMR in a country like Ghana. Following the World Health Organization (WHO) strategy on fighting AMR [2], Ghana developed a comprehensive national action plan (NAP) on AMR for 2017–2021. However, not only is the quality of the NAP important, but even more so, is that of its implementation.

### 1.1. AMR-Related Politics and the Development of the NAP

Regulations and programmes relevant for fighting AMR in Ghana date as far back to 1998 when Ghana adopted the Rational Use of Medicines Programme (RUM), which complemented the “Essential Drugs List and National Formulary with Therapeutic Guidelines” (existing and periodically renewed since 1988), which was separately published later as the Standard Treatment Guidelines and Essential Medicines List. Since 2000, the Ghana National Drugs Programme (GNDP), a pharmaceutical policy unit of the Ministry of Health (MoH), has controlled these activities [3]. In 2013, the Health Professions Regulatory Body Act regulated the permission to prescribe medicines.

RUM included the formation of Drugs and Therapeutic Committees (DTCs) in the majority of the teaching, regional and district hospitals. Their tasks focussed on the monitoring of antibiotic use in their facilities. The RUM programme contributed to a decrease in antibiotic prescription at the level of Outpatient Departments (OPDs) from 56.3% in 1999 to 41.4% in 2012. Nevertheless, between 2002 and 2004, a first nationwide surveillance of antibiotic resistance in Ghana observed a high percentage of resistance for tetracycline, cotrimoxazole and ampicillin, which are frequently used antibiotics in Ghana [4]. However, the large incidence of self-medication and the wide availability of antibiotics, including inappropriate prescriptions in many health facilities, have resulted in a continuing overuse of antibiotics [5,6]. Preliminary findings from a point prevalence survey in two peri-urban health facilities show about 60% of inpatients were on at least one antimicrobial medication. There has been extensive microbiological research on antibiotic/antimicrobial resistance in Ghana since the late 2000s [7].

The discourse on AMR, which led to the launch of the Ghanaian NAP on 11t April 2018, was initiated by a Danish research cooperation project on “Antibiotic Drug use, Monitoring and Evaluation of Resistance in Ghana—A research capacity building project (ADMER)”, which operated from 2009 through 2015 in partnership with the University of Ghana Medical School (UGMS) in Accra and the Kwame Nkrumah University of Science and Technology (KNUST) in Kumasi [8]. Furthermore, Action on Antibiotic Resistance (ReACT) Africa has accompanied the networking activity among Ghanaian stakeholders since 2011 and further on, the implementation process for the Ghanaian NAP. ReACT is an extended network funded primarily by the Swedish International Development Cooperation Agency (SIDA) [9].

A milestone event for the Ghanaian discourse on fighting AMR was the establishment of the Ghana National Policy Platform for AMR (NPAR) in February 2011 at a stakeholder meeting involving local institutions from the health sector, academia, donors and research networks supported by ADMER and ReACT. In the following years members from various fields joint the AMR platform: non-governmental organisations (NGOs), industry, media, health professional bodies, regulators, the animal/veterinary field and the environmental sector. From 2015 to 2019, about 190 members from the aforementioned institutions actively participated in the NPAR work [10]. A recent article by Koduah et al. [10] provides detailed information on the role the NPAR plays in sustaining the national AMR agenda. These include supporting and sharing research findings, strengthening AMR awareness creation, supporting training and advocacy, gaining and maintaining political support and supporting the implementation of the national action plan on AMR. As a further achievement, the authors stressed that Ghana host the second Global Call to Action on AMR meeting in 2019 as an outcome of the sustained AMR agenda and progress towards curbing the AMR challenge. The platform has since developed into an extended network and has become the central backbone of cooperation between national actors. The multidisciplinary background of the members on the NPAR also set the scene for integrating the “One Health” approach into the national discourse on AMR [11].

### 1.2. NAP: Its Final Shape

In 2017, the Ministry of Health (MoH) first published a policy paper on the situation concerning AMR in Ghana, the policy goals and the strategies on surveillance, research, reducing the incidence of infection and optimising “the use of antimicrobial agents in humans, aquaculture, plant production and in animal health in the ‘One Health’ approach.” This began the groundwork for creating an “enabling environment for sustainable investment that takes account of the needs of Ghana, and increase investment in new medicines, diagnostic tools, vaccines and other interventions” to be developed [12].

Shortly thereafter, the NAP was published after cabinet approval in December 2017. In April 2018 Ghana’s President officially launched the National Action Plan together with the Tripartite Bodies in the fight against AMR (WHO, the Food and Agriculture Organization of the United Nations (FAO) and the World Organization of Animal Health (OIE)) represented by the WHO country representative, Dr. Owen Kaluwa [13]. 

The NAP is a complex document including a timeline for the implementation of five strategic objectives and sub-objectives (2017–2021), an operational plan specifying the implementing institutions and collaborators for each of these items, a detailed budget plan (cost of each item and expected source of funding) and a monitoring and evaluation framework. The NAP basically follows guidelines by the tripartite institutions [14].

Table 1 represents the overview page for the monitoring and evaluation framework. It includes the title of the five AMR goals and formulates outcome statements for each goal [12], which are supplemented by long lists of outcome indicators and a detailed compendium of data sheets to be completed during the implementation.

### 1.3. AMR Policy Structure in Ghana

In its first part, the NAP describes an AMR country level governance framework which can be seen as the policy backbone for monitoring and evaluating the implementation of the AMR goals stated in Table 1. An “AMR inter-ministerial committee”, “made up of the Ministers of the affected Ministries to help drive the AMR agenda at the highest level of governance” [12] (p. 6f) is located on top of the governance framework; at the core, the NPAR (“multi-stakeholder platform”) is situated with its linkages to other stakeholders, foreign donors and partners and the AMR Secretariat. Implementing agencies are coordinating the whole network as illustrated in Figure 1.

We should consider that there are a large number of stakeholders belonging to quite different groups of actors (see above, section “AMR-related politics and the development of the NAP”). Figure 2, Ghana AMR Policy Process Flow, by Priscillia Nortey (Member of the NPAR) offers a comprehensive overview on the policy process for AMR policy and thus can be useful for pointing to research questions needing to be clarified by further research [15]. It presents a flow diagram on the policy process which led to the Ghana NAP on AMR, and in a second loop, links it to the policy implementation process. In her presentation, Nortey points out: “By implementing the NAP, ministries, agencies and departments (MDAs) would be implementing the AMR policy for Ghana. Guidelines have been developed to assist MDAs to mainstream the AMR NAP into their POWs [programmes of work; medium term strategies]”. Nortey’s “AMR Policy Process Flow”, presented a year after the completion of the NAP, includes a process for re-negotiating the NAP, but does not indicate an analysis of progress and setbacks in various fields of the implementation during the phase until 2021, nor is possible feedback on specific problems included.

In our desk research on developments up until early 2021, we found a few concrete indications on how these links between the NAP approved on the national government level and concrete implementation activities by stakeholders in specific sectors and regions have progressed. It is likely that our internet-based impression of a rather limited presence of AMR and the NAP in general Ghanaian politics is incomplete, and we should aim to complement them with more in-depth information through interviews. This also refers to what Nortey calls the “Civil Society arm of Action”; in the internet sources on AMR and the NAP, “classical” civil society organisations are hardly visible. However, George Hedidor, representative of WHO, mentions in a comment to the authors of this study (e-mail from August 2021), that the Ghana Coalition of NGOs in Health was represented throughout the process.

### 1.4. Overall Research Objectives

A lack of transparency on political processes resulting from limited publicly available information has been shown to exist in nearly all national action plans on AMR in 15 African countries [16]. Country studies on Tanzania [17] and Cameroon [18] lead to the result that implementation by far has not achieved stated goals. In general, evidence for successful and comprehensive national AMR policies and programmes in low- and middle-income countries (LMICs) to date is scarce, and a number of causes such as fragmented health systems, poor governance systems and corruption, lack of financial resources, have been identified [19,20,21].

As has been shown above, Ghana can look back at an extensive discussion on AMR issues and a rather realistic assessment of the situation at the onset of its NAP on AMR [6,22,23,24]. The NAP has defined ambitious and extremely differentiated goals and procedures. Nevertheless, until mid-2020, desk research delivered hardly any accounts about concrete progress along the lines of the NAP, leaving the impression that despite all previous attention to AMR, the quality of implementation of the NAP is relatively low, comparable to that of many other LMICs. The aim of this empirical research is to assess the progress and impact of the AMR NAP in Ghana and identify ongoing NAP related activities which are not reflected in publicly available information online. We will also evaluate the accomplishments and shortcomings regarding the implementation of the NAP. 

## 2. Methods

### 2.1. Study Design

An exploratory research design using a qualitative study approach was used to explore the political processes and governance of the AMR NAP in Ghana. Due to limited financial resources for this project, we could not use more complex frameworks of governance analysis, such as that proposed by Anderson et al. [25]. Key informant interviews and a review of articles, policy documents, reports and other grey literature were conducted to expound on the process leading to the development and implementation of the NAP. 

### 2.2. Study Participants

We identified various experts and key informants from various stakeholder groups comprising government institutions, academic bodies, private and civil society organisations, international development organisations and donor agencies who actively contributed in the discourse leading to the development and implementation of the NAP. The participants were selected based on their previous contribution or active involvement in the AMR discourse, policy dialogue and implementation of the NAP. 

### 2.3. Data Collection

An extensive online literature research was conducted using the search terms “antimicrobial resistance” OR “antibiotic resistance” OR “AMR” and “NAP” OR “national action plan” and Ghana. Additionally, a snowball technique was used to further identify relevant information mentioned in documents or articles found from the online literature search. Based on materials published between 2016 and 2018 summarising the state of affairs at the onset of the NAP and research on press coverage and project activities, we developed a set of research questions and identified potential interview partners for key informant interviews. A semi-structured interview guide was developed covering the development and implementation of the NAP, AMR governance and public discourse, AMR training and advocacy and the contributions by Ghana in the international fight against AMR. The interview guides were tailored to each of the interviewed experts. Due to travel restrictions caused by COVID-19, all but one interview was conducted via online teleconference. Out of the of 25 experts and key informants who were identified and contacted, 13 were available and consented for the interviews (Table 2). The interviews were carried out between 23 February and 29 March 2021. Each interview lasted an average of 90 min. All the discussions were audio-recorded and transcribed. Though it was not possible to organise interviews with all the experts proposed, authors of some of the most important contributions discussed above and representatives from the most important stakeholders of the NAP/AMR process could be included; regrettably, no interviews with experts from the media or representative from the Planning, Monitoring and Evaluation (PPME) directorate could be arranged. Table 2 presents an anonymised list of the experts interviewed and their institutional backgrounds.

### 2.4. Data Analysis

The thematic analysis approach was used to identify and analyse emerging patterns in the data. The themes were developed from both the interview guide and from the narratives of the participants. All the transcripts were read twice to develop a framework of the emerging themes. The participants’ narratives were triangulated with each other and the available literature on the development, implementation and governance of the AMR NAP to understand the correctness and the significance of the emerging discourse.

## 3. Results

The review of the NAP and attempted corroboration of the implementation status with additional data sources revealed various gaps in information. The NAP analysis based on the desk review helped to generate a list of emerging themes which required information from key informants through interviews. This led to 10 basic research issues out of the Ghanaian discourse on AMR and to formulate corresponding research questions to structure the expert interviews. Information from more recent materials, which were gathered later and include aspects of evaluating the implementation of the NAP, are integrated into the report of the interview results, organised corresponding to the respective questions. 

### 3.1. Emerging Areas of Research

The review of the NAP and attempted corroboration of the implementation status with additional data sources revealed various gaps and provided the basis for the 10 major issues which we explored through the expert interviews (see Appendix A for more background information). 

During the process for developing the NAP (and even before), a considerable body of research on the prevalence of AMR related to specific substances and in specific locations in Ghana had been published [6,26], including overviews on nationwide surveillance of antibiotic resistance [27]. In the context of the NAP, a more precise nation-wide AMR surveillance (continuous monitoring the prevalence of AMR as well as the consumption of anti-microbials) is essential. Nevertheless, Ghana did not deliver data for the 2020 and 2021 WHO Global Antimicrobial Resistance and Use Surveillance System (GLASS) Reports [28]. 

Available documents and analyses consider One Health as a concerted approach to addressing conflicts among stakeholders in the fight against AMR. In various papers, however, it has been argued that structures to target “One Health” approaches in Ghana need improvement within the veterinary services [6,23,29]. This is also pointed out in a study on “Antimicrobial drug usage and poultry production” [29].

The paths for procurement of medicines, legal and illegal, are essential for controlling access to antibiotics. It has to be taken into account that even legal channels of procurement might not necessarily comply with all regulations (e.g., lack of surveillance, corruption), that some regulations might be contradictory [6] and that counterfeit and substandard medicines frequently enter the supply chain in many countries. A study on the use of antibiotics at a market place in Kumasi [30] shows that women in particular purchase antibiotics from non-pharmacy outlets and market peddlers. More information on the efforts of the Food and Drugs Authority (FDA) is needed.

Central to understanding governance structures and their shortcomings is more information on the political processes for transforming the concepts of the NAP into concrete regulations and creating the conditions to enforce them. According to the NAP, the MoH set up an AMR Secretariat, which convenes quarterly AMR Platform meetings, and other governance links. Furthermore, the Policy Planning, Monitoring and Evaluation Department at the MoH plays a crucial role. Online information on the concrete operation of these institutions remains scarce.

The implementation of the NAP requires substantial financial resources. The total budget (for 2017 to 2021) was projected to be USD 21,276,047.93. While there is a very detailed financial plan “costing” nearly all 150 NAP elements, the funding structure for the NAP remains unclear since for the majority of the NAP budget items listed, the sources of funding are indicated as rather generic categories such as GOG (Government of Ghana), DPs (Development Partners), Corporate Institutions and NGOs. In general, no specific ministry budgets or organisations are named. When reviewing the Budget Statement and Economic Policy of the Government of Ghana for the 2019 Financial Year [31], the terms “NAP” or “AMR” or “surveillance” (related to health) do not appear. These findings are also reflected by a recent study on transparency of AMR NAPs in African countries, which found that budget transparency was very low [16].

The WHO Country Level Report [23] focusses on the issue of “getting AMR into existing programs” (note the main title: “Resource mobilisation for AMR”). Certainly, there is a good chance to mobilise external funds for many activities to be developed with the implementation of the NAP. Strategies for that are proposed within the WHO report and in some other papers [23,32]. This implies that international partners have to play an important role for the development and implementation of the NAP. From the governance perspective, the AMR Secretariat and the AMR Platform should play a crucial role, but that is not clear from the available documents [32]. 

Until 2019, the Fleming Fund Ghana Country Grant [27] was the only large cooperation project, which supported the comprehensive implementation of the NAP. It had a volume of USD 2.352 million, which amounts to 11.05% of the total budget for the NAP. The grant objectives are summarised in three points (see also: Fleming Fund, website [33]):○Establish a well-functioning One Health governance structure for AMR and AMU surveillance;○Establish a government led system of collecting, collating, analysing, reporting and disseminating AMR and AMU data on national and international platforms in alignment with the requirements of the WHO GLASS;○Strengthen AMR and AMU surveillance in animals.

Further important issues that require attention in the evaluation of the NAP implementation are AMR-related capacity-building in academic institutions, the level of awareness in Ghanaian society (see [34]) and the role of Ghana in the international discourse on AMR, for example, its role in the Global Call to Action on AMR and the Ghanaian approach to the formation of a national working group on antibiotic resistance (the NPAR) [10,35].

Finally, the review of the literature left some doubts about efforts on the monitoring and evaluation of the NAP (2017–2021). As early as January 2017, the WHO presented a detailed concept for Monitoring and Evaluation of the Global Action Plan on Antimicrobial Resistance [36]. So far, however, no comprehensive monitoring and evaluation report on the Ghanaian NAP has been presented. In August 2019 a short press release titled “Ghana’s National Action Plan on AMR on Course” was published by the Ghanaian government, stating among others, the launch of the AMR Policy and NAP, and the first Interministerial Committee meeting to be held on 21 March 2019 to sensitise implementing ministries and agencies on the AMR country actions. Thus, the “highest decision-making body with oversight responsibility for the implementation of the interventions on AMR” started one year after the Ghanaian president had officially launched the NAP (which formally had already started in 2017) [37].

### 3.2. Results from Interviews and Most Recent Information

#### 3.2.1. Incomplete Implementation of Surveillance Systems

Without effective surveillance of antimicrobial use (AMU) as well as of AMR prevalence, control of AMR is impossible. Some of the experts interviewed made reference to a WHO supported pilot study on AMU in 2015. However, this has so far not yielded the needed evidence due to the stagnation of the project. 

Most experts refer to the importance of the Ghana Country Grant by the Fleming Fund (FF) [29], which was seen as a promising starting point for the development of a fully integrated surveillance system at the national level, including AMU and AMR prevalence. This goal, however, had not been reached at the end of the running time of the grant (December 2018–May 2020). According to experts, three problems persisted, mostly related to the lack of necessary infrastructure: insufficient laboratories, digitalisation and electronic systems (e.g., prescriptions not digitalised (R8)); lack of coordination between telecommunication systems used by different stakeholders (R11); lack of research on the diversity of bacteria from hospital environments [38]). To complete the work, a Country Grant 2 was provided for the period between December 2020 and July 2021 [39]. Respondents considered it challenging that so far Ghana has not been able to contribute data to the annual GLASS reports. 

Expert R2 summarises the experience on the surveillance of AMU.

“…when you come to antimicrobial consumption, …there was a pilot… done, through the support of WHO Ghana, to get the data from various selected health facilities. And so, that is one of the first pilots that we’ve done on how to capture antimicrobial consumption data in Ghana. And then also under the Fleming Fund, we started a process to get the data on import, local production, local export of pharmaceuticals; the drugs, so that we can know the quantity of antimicrobials consumed in a year in the country. And that process was also started but it has been put on hold. Maybe for second phase of the FF, it will be completed. So, there’s a process in place.”

In addition, a project organised by DTU (Danmarks Tekniske Universitet) was mentioned, which analyses resistance patterns of bacteria in samples from different countries, among them Ghana, funded by the DTU’s IGF (internally generated fund) and supported by WHO and the Ghanaian Water Research Institute (R4, R5).

Based on initiatives from universities in cooperation with external partners, a number of point prevalence studies have been performed and/or started at: (a) KNUST Hospital (UHS), Agogo Presbyterian Hospital (APH) and Ejisu Government hospital (EGH) [40]; (b) Keta Municipal Hospital (KMH) and Ghana Police Hospital (GPH) [41]; (c) Ho Teaching Hospital [42]; (d) Point Prevalence Survey at Agogo Hospital and at St. Francis Xavier Hospital at Assin Fosu [personal information from BNITM and KCCR].

Most surveillance activities have been related to human health; as they constitute mostly separate projects, no comprehensive results for Ghana can be provided. Furthermore, the situation of informal access without prescription in most parts of Ghana (see also RQ 3) impedes surveillance of antimicrobial use. Experts also refer to research related to the animal health sector and the environment—“consumption data…in the veterinary services is scanty”, similarly, in the case of data from environmental surveillance (R5).

#### 3.2.2. Importance of the One Health Concept

The One Health concept (OH) was introduced during the last phase of the work on the NAP. A publication from 2018 on One Health integration in veterinarian and zoonotic disease management states: “… Ghana does not have an organization, government department, or official plan with a clear mandate to pursue OH” [43] (p. 3). Thus, the NAP is probably the first national document/plan, in which the OH played a significant role. R13 confirmed statements by Yevutsey et al. [6] and WHO [23] that there were no shared efforts among users of antibiotics. He referred to the formation of the “One Health Committee” and saw some progress:

“…now we share our information at the CC [AMR coordinating committee] meeting every quarter. … we know what is happening in the human health sector, we know what is happening in the environment. This is what we are doing now as a committee at the AMR level”(R13)

Other respondents also stressed the integrative impact of OH: “…the NAP is developed in one health. So that is one of the major successes….the WHO Ghana Office, the FAO and then OIE… [are] working with the platform” (R1; similar: R9; R6).

While in the human health sector there is no direct legal conflict between policies to control AMR and economic interests (besides issues of finance), the situation is different in the field of animal farming, where “rearing animals for food and jobs”, using “European style” raising of animals has been supported by development politics (in particular in the field of intensive poultry farming, characterised by overmedication). “We are also working with the OIE and all the others, FAO, we also trained poultry farmers in Dormaa Ahenkro” (R13; R6); this is a model field school to stop the “inappropriate use of antibiotics” in poultry production [44].

#### 3.2.3. Insufficient Enforcement of Regulations for Procurement and Dispensing

In principle, the legal paths for procurement of medicines and the control of the access to antimicrobials are strictly controlled by several institutions (border control, Ghana Food and Drugs Authority (FDA), MoH pharmacy) and various laws and regulations focussing on the control of quality of medicines and preventing the entry and sale of fake medicines (this refers to the WHO term of “Substandard and Falsified (SF) Medical Products”). Respondents, however, pointed out that there was no effective enforcement of these regulations. 

“…the pharmaceutical- the health professional regulatory act, talks about what we’re supposed to do when a medicine is dispensed in Ghana. Such implementations are not ongoing. There’s no national intervention to ensure that …the recommendation from the law is being implemented. So, then we still have, I’ll say, lack of control… over the dispensing of drugs in the country”.(R2)

The lack of control coincides with income opportunities for groups who illegally procure antibiotics either by effectively using the loopholes of the local pharmaceutical markets, or by various forms of smuggling.

“There may be promoters of resistance in the society by their activities. They have no knowledge about the antimicrobial that they sell around. All they want is money. So, any sickness mentioned, they’ll give you the antibiotics. Whether it will kill the bacteria, whether the sickness is caused by a bacterial infection, they don’t care”.(R2)

“[Smugglers] pass through the unapproved routes, the borders. Yes. So, they always smuggle antibiotics into the country. Whether they are safe or they are efficacious or not, is another issue” (R2). “It’s a whole cartel that is working, a kind of a global menace. …These drugs come through illegal routes … Let’s stick to antibiotics. Antibiotics will need to be stored appropriately… to retain its efficacy. So, I go out I buy some boxes of amoxicillin injection or a few parts of …meropenem or, aminoglycosides…Because they’re not bulky, so if I have a suitcase, I can buy let’s say 10, 20 right, then I come and supply it…. I can buy it at a cheaper cost than the ones that have been imported by those who have registered the products to use. So, we cannot be sure of the supply chain integrity of such products”.(R1)

One of the main issues constitutes the control of counterfeit medicines. Bright Simons, a renowned Ghanaian innovator, has developed a widely used system to use mobile phone-based technology to check for counterfeit medicines, mPedigree. However, they were not successful in rolling it out in Ghana: 

“we’ve been trying for… more than a decade. …Finally, it became quite clear that we will not be able to make any headway in Ghana. So, we left Ghana and we spent time in Nigeria and Kenya and other places. And yes, in those countries, particularly in Nigeria which is the most advanced, the solution and others like it is used for controlling authenticity.” He further stressed: “… I see the traceability and supply chain solution as the base on which you can build other higher value adding solutions some of which include antimicrobial resistance, surveillance and monitoring systems”.(R12)

Other respondents confirmed this, but added that the MoH is now using another medicine traceability strategy, which is currently being developed (R2, also R6). Nevertheless, the situation described above seems to prevail. 

#### 3.2.4. Governance Structures for Implementing the NAP (Institutions and Regulations)

Concerning core capacities (surveillance, links between human health, animal health and environmental issues focussed through “One Health”, problems of access to antimicrobial medicines), progress and shortcomings to the implementation of the NAP alike have been stressed by the experts, drawing attention to the system for governance of fighting AMR. While the “Ghana AMR Policy Process Flow” (Figure 2) gives an impression of a well-organised policy process, it remains abstract with respect to the central points of implementation and evaluation. An expert from the WHO country office characterised strengths and weaknesses of AMR/NAP governance:

“Well, health systems in general in Ghana, is not at its peak... if you take one aspect, governance alone is good on paper but practically it’s not so strong”.(R9)

“When you look at the NAP or the policy, there is a governance structure [description of the governance framework presented in Figure 1]…the secretariat coordinates all the activities, and the inter-ministerial body is the highest decision-making body, and the platform is like a clearing house where everything is brought there... decisions are made, directives are given and we go and work. That is the governance structure. In terms of reporting, the secretariat is supposed to demand reports from all agencies including the GHS and collate that into one as Ghana report. That is how the system works. … But... close to 60, 65 percent of activities in the NAP are still hanging... not started at all because of funding issues so you can see that if they have not started you will not get any report” (R9). The same respondent added (after a long explanation on the ubiquitous access to antibiotics): “So, there are laws but the *enforcement is weak.* It is a system-wide problem. It is not just about the regulatory agencies. I keep saying that it is more of access and equity …” (referring to difficult access to pharmacies in remote regions).(R9, also R1)

These quotes indicate that there has been a continuous involvement by a significant number of stakeholders and communication among them is working, i.e., there is a basic functioning governance structure (also R2). On the other hand, implementation is essentially hampered due to a lack of funding (R1; R4) and, presumably related to this, a lack of enforcement of existing regulations.

Work on regulation and legislation is in fact going on:

“…we were able to take up pieces of legislation on AMR from all sectors. From environment, from health, from Agric; if it comes to Agric, from animal side, from crop side, from fishery side. So, we were able to pick those pieces, put them together as AMR legislation for Ghana… Now, we have gotten to a stage where we are working on passing … the revised version into law and if that is completed, then we can use that law to implement most of the AMR activities because everything should be backed by law… So, with time, as and when we get this recommendation passed into law, the enforcement will be good. But as at now, we haven’t achieved what we want to achieve in that area”. (R4)

While there was a general consent that enforcement in the regulation of access to antibiotics was low, there was some optimism about the chances to improve the legal foundation of AMR-related policies (provided there are more financial resources).

“…let’s push budget there and be able to implement them. So, now, the policies are being used in the various organizations for the implementation of the AMR activities, and another thing we did was to help ministry departments and agencies to mainstream the NAP into the medium-term development plans, so that AMR activities become part of their organizational plans for implementation”. (R4)

An expert from the MoH responded on a question about the weak enforcement of existing regulations: 

“I think the challenge is very much there and what we are trying to do is to leverage on certain existing laws which are luckily been reviewed now, so that we can leverage on these reviews to enforce the law. So that problem still persists, so this is how we intend to address the problem”.(R6)

Responses to questions around governance indicate that a significant number of papers were prepared by the various organisations and institutions for the quarterly meetings of the Secretariat (R4). There are, however, no webpages of the AMR Secretariat and the NPAR, which might publish such papers. The MoH webpage (https://www.moh.gov.gh/ghana-health-service/) (accessed on 1 May 2022) provides links to 26 health agencies, but not to the AMR secretariat. This lack of publicly available information suggests a low transparency on the implementation process for the NAP.

#### 3.2.5. Mobilizing Financial Means for Implementing NAP

The lack of finance was frequently mentioned as one of the most critical aspects for the NAP implementation. Most of the experts saw the lack of government funding as the central problem, as expressed in the following statements:

“The key problem I see is financial. When you look at the NAP activities, all the institutions involved have the capacities to implement the activities as stated in the NAP, so if we get the funding the NAP can be implemented according to the plan”.(R10)

“All the researchers on the platform have redirected our research into AMR related issues, but I have not seen a kind of dedicated budget of these ministries. If there are dedicated budgets…perhaps there would have been more frequent engagements with our partners and stakeholders…towards the implementation”.(R1)

“The NAP was supposed to be integrated into the sector ministries budget… as we speak now, I can’t tell whether it’s been done yet. So again, the reliance has been on external funding… if we should really want things to work then we should be looking at budgeting for it and getting it implemented… I don’t think the sector ministries have them in their budget”.(R11)

The FAO coordinator, however, proposed a more differentiated way of thinking on this issue: 

“You know when the AMR policy and the National Action Plan were ready, it was the President himself that launched those documents at the International Conference Center, and that day, he urged the Minister of Finance to support the implementation of AMR and the national action plan. So that is a plus. Soon after that, the Minister of Health in a very good initiative brought the Global Call to Action on AMR to Ghana. …And that brought international AMR stakeholders to Ghana. Ghana was given money from the UK Government for AMR implementation…. And again, I always tell at our meetings, …, yes, government is supporting AMR in so many ways that we don’t see directly. For example, …infection prevention activity is purely AMR activity…Likewise, …the Veterinary Services, they’ve also got budgets for infection prevention activities. The vaccinations that we have been doing all around every year is part of AMR, is pure AMR activity. If you do vaccination you… prevent the development of diseases, then you don’t use antimicrobials. So, in this way, I think the government is mobilizing funds for the support of the AMR. What…we haven’t seen is actively… bringing in money … with the title that this money is going for AMR. But there are so many funds that are coming to the system for implementation of development plans. …if we are able to quantify those things, then we can see that yes, there’s enough budget allocation from the government for AMR”.(R4)

It has to be seen, however, whether these cooperations become functional if (financial) government commitment on fighting AMR is not made explicit but remains to a large extent “hidden” (as explained by R4).

“Some of these international communities, they want to see government commitment. But in terms of government commitment, I haven’t seen that level of commitment. May be the environment sector’s implementation is strengthened because UNEP is now involved”.(R10)

#### 3.2.6. Mechanisms of Mobilising International Partners

Mobilising international partners has been successful during the development of the NAP and should have been a priority considering the financial challenges highlighted by the respondents. In fact, the general complaints about the lack of finance point to a situation where external finance is also not amply available. One aspect to be considered is the lack of an integrated management of international cooperation; summarising quotes from the interview with the deputy chair of the NPAR, reference was made to a considerable number of partners having cooperated with Ghana since the onset of the AMR discourse leading to the NAP (DANIDA, SIDA, ReACT, FAO, WHO, Wellcome, Fleming Fund, “some Germans”, etc.). He also referred to an important problem inherent in capacity building particularly through international cooperation, the phenomenon of brain drain:

“I think, it would be great to even bring all of the supporting institutions to work together synergistically, some of these are very disjointed. So it’s only those of us who are working with the platform, we get to know about them and we inform… and they try to embrace or try to get information from … their [partners]… and then how relevant it is towards the NAP implementation”.(R1)

“But then we have had these challenges, where even at a point in time, you know, equipment that are needed in the field to work and generate the needed data has not still gone down to the field, in spite of the fact that capacity of the practitioners has been developed [reference to the interruption of the Fleming Fund project on surveillance, see above]. There are also fears that even once [human] capacity is developed, and they get some lucrative job, they may leave so there may be the need for retraining, once we’re able to get things in place. But I agree that there has to be, … somebody who is looking at; the … AMR Secretariat is being actually tasked, … but I don’t know if they have the powers to work with [international organisations]”.(R1)

A detail was added by R2: “There is a desk in the Ministry that is responsible for that…at the PPME directorate.” In fact, this directorate comprises a Resource Mobilization Unit with the following tasks: “The Unit develops and reviews the financial strategies or options for policies/programmes and projects of the Ministry. It leads in the sourcing, monitoring and coordination of external funding for healthcare delivery, to include sourcing external funding for the procurement of health commodities, services and works”. There is, however, no further link to enter the PPME directorate [45].

Many respondents referred to the central role of the tripartite organisations (and also UNEP) in coordination and in supporting ongoing activities important to combat AMR, that is antimicrobial stewardship (AMS): 

“We cannot implement AMR activities without international cooperation; most of the funding comes from WHO and FAO. They supported in developing the National Action Plan. Coordination at the national level is spearheaded by WHO. So, if WHO is going to undertake some activity, they consult FAO, OIE and now UNEP. MoH hosts the AMR secretariat and they organize the platform meetings”.(R10)

“I’ll say that the Tripartite in general actively supported the country to come out with the AMR policy and then the NAP and of course, we are still supporting them to implement some aspects of it… We basically provide technical support and some funding to ensure activities are implemented. …Collaborations with FAO and OIE are excellent, I know, there are AMS activities ongoing by various agencies (Kintampo Health Research Center, KCCR, KNUST). What I can say too is that there is a project coming... the Multi-Partner Trust Fund Project which seeks to pool these agencies together in One Health. Out of that there are crosscutting activities which are supposed to be implemented across the three agencies and that project would hopefully begin somewhere April if all things are put in perspective..(R9)

The role of CSOs in international cooperation remains unclear; in some cases, they might be included just as a fig leaf, as in the case of “Health for Future Generations” in the Fleming Fund Grant: 

“We don’t have any role, we were part of the development; our organization contributed, we supported with a letter, everything but when the grant came, we are not involved so we don’t know what exactly they are doing…Our social media team also support creating […] a lot of information but we don’t have any project that is funded by anybody. …For now… ReAct Africa is doing a lot of awareness work”. (R7)

As these quotes show, also among experts who have been involved in international cooperation, there was considerable insecurity about a central management of cooperation projects within the Ghanaian government. Two other new cooperation projects were mentioned in the interviews, which are also referred to in the literature: “SORT IT (the Structural Operational Research Training financed by the Special Programme for Research and Training in Tropical Diseases (TDR) and WHO) supports countries to build sustainable capacity to conduct and publish operational research and use the evidence for informed decision making to improve public health. The goal is to make countries “data rich, information rich and action rich’” [46] (quote from Section 4.2.3 of the WCO Ghana Annual report, no pagination). ReACT supports training as well (R9).

Supported by the Global Fund and USAID, an important element of modernising the logistics within the Ghanaian health system has been developed:

“With the implementation of the Ghana Integrated Logistics Management Information System (GhILMIS) which is being rolled out nationally, we are hoping that we will be able to obtain consumption data moving forward. This roll out is ongoing…, we are planning a meeting where we will ask each region and even each district to nominate a focal AMR person, who will be tasked to make sure the consumption data is collected and reported nationally to feed into the [..] systems and eventually into the WHO GLASS”.(R6)

According to the Ghanaian Ministry of Health “expected that by November 2020, all health centres and “functional” CHPS [Community Based Health Planning and Services] would have been successfully signed onto the system fully” [47,48]. Internet information available in early 2022 [Google, search term: “Ghana Integrated Logistics Management Information System”] does not give clear information on how far the development of GhILMIS has in fact proceeded. It is certainly an ongoing process, progressing through various Master Plans [49] and various international cooperation projects (most important with the Global Fund and with USAID). 

#### 3.2.7. AMR-Related Teaching, Training and Research in Academic Institutions

Respondents agreed that AMR had not been integrated into university teaching as a specific module, but that it played a significant role in many courses, basically in infectious disease management. It was revealed that many students showed interest in the topic with some Master’s and PhD dissertations being written in this field. However, the perceptions of MoH impact on university programmes differed among respondents.

[Modules of AMR and AMS?] “No. Currently there are not. … But you know, universities are independent, so we are suggesting but universities have to take time to buy it into their own system…I would not say that there’s no strong interest on AMR in the universities. I suspect that because the NAP has come and has to be integrated into the curricula of the universities, it is now something that they are trying to put in place, because they already had their own programme before and I think that curriculum is not changed every year in the universities”.(R13)

The respondents also referred to a number of theses and publications carried out at the universities. Furthermore, it should be pointed out that AMR and AMS is included in teaching on infectious disease management:

“…in pharmacy, we have actually incorporated antimicrobial stewardship and the role of the pharmacists in AMR containment, at different levels, graduate levels, undergraduate levels and all of that is actually part of the curricula for infectious disease management. But I think we should go beyond that, we should be able to see that in the curricula of Veterinary Medicine, we should see that in the curricula of School of Medicine and Dentistry, we should see that in the curricula of Laboratory Science, you know, all of that so I think we need to move on”.(R1; similar R3)

[Interest among students?]” Yes, huge, you know, … very, very, very huge. A lot of them even take up mini projects, alright, on that …. So, if what is happening in pharmacy can also be replicated, you know, medical school, nursing, at least in their final years or a year to the final year, if they are actually taught about some of these issues, I think that would really be great. But I don’t know which institution or organisation will lead that kind of advocacy”.(R1)

[Oversight by the secretariat] “You know the NAP recommended, that we should have AMR issues included in the curriculum of the teaching of the universities… So, we know that one university, KNUST, started and they’ve done some inclusion already. But the others, we’re yet to find out from them whether they’ve included issues of AMR in their curriculum…. From time to time, the secretariat goes through some of the training institutions, to create awareness…”.(R2)

In addition, it should be mentioned that WHO and FAO are carrying out a number of training programs on specific issues, such as the SORT IT mentioned above, or training for professionals in agriculture and fisheries, a “Private Sector Engagement Workshop on Antimicrobial Use and Resistance” and programmes for journalists [47].

#### 3.2.8. Awareness Building and the Role of Civil Society (Societal Embedding)

While academic institutions have provided an important contribution to awareness building on the professional level, AMR Goal 1 rightly goes beyond that to increase awareness in the general population. A respondent from the Environmental Protection Agency replied: 

“It will interest you to know that I was not even aware of this AMR thing, until I joined the whole AMR discussion. I think in 2018 there was the WAAW week, they held a programme at the Ministry of Health, that was the first time I participated in it and I understood the whole AMR discussion. So, you see it means that a lot of people are not aware of this whole AMR issue. And I think the awareness creation has been limiting for a long time. Apart from the WAAW week I don’t see any rigorous awareness on this whole AMR thing”. (R10)

“At least every November onwards [WAAW] something should happen and people are aware. … But I can tell you a lot of plans are put on paper but our basic challenge has been funding because people are prepared to work…. People don’t put money into awareness issues because the deliverables are not tangible. I’ll say that gradually the awareness is catching up”.(R9)

“I think the problem with the media is …to do more of the training. I think they have to understand the AMR issues. … even when you have spoken to them about it, when they are reporting, sometimes you realise that the substance in what you shared with them is not what actually they are reporting… When we are doing World Antimicrobial Awareness Week we are going to see more of the media, … we see people going there, buying airtime, or maybe as part of their corporate social responsibility, giving them the platform to talk about AMR issues. … And I think that, you know, we should also work very hard to make the media guys very strong advocates, once they understand the issues, and they make noise about it, and I think that’s one sure way of ensuring massively a real change”.(R1)

[Around 2015] “… the awareness level was measured to be so low, even among the health workers, let alone the general public. The Kintampo Health Research Centre for instance, did the KAPP [Knowledge, attitude and perception/practices) [50] research on awareness creation—Awareness on antimicrobial resistance among the health workers… Objective one [in the NAP] … is awareness creation; every year we do this awareness creation, CSOs are doing the awareness creation, but nobody has gone back to find out what is the level of awareness among the general public”.(R2)

CSOs in Ghana are working to improve awareness on AMR, but according to a respondent from “Hope for Future Generations”, with limited success and without much financial support:

“So, the regions where we [HfFG] have projects actively working, yes. …So, if you ask me about awareness, I think its limited, I will say maybe 0.5, 0.3% of people are aware, maybe listening to TV and radio. You know, when we are celebrating AMR week… I don’t think people are aware because when you talk to people … they become surprised, so we are not doing much when it comes to awareness creation.… Civil society can do more than what they are doing because that is our field; because working with community members, communicating in their language is our work. So, I think we need to just really appreciate, there is a gap that we have, big gap that we must address…you know we work a lot in the communities. When we are meeting mothers for our maternal health, we talk about it, but receiving money purposely for this, no.… Communication [with the NPAR] was very active when we set up the platform. Now the communication, I don’t see it very active but periodically you will see some information on the platform”.(R7)

The statements from our respondents confirm the assumption based on internet research, that AMR is not presented as a priority topic in the Ghanaian mass media and in the general public. The impact of the WAAW remains limited, as long as AMR remains hidden behind other health topics such as malaria, and recently, COVID-19. 

#### 3.2.9. Ghana’s AMR Politics in the International Context

Among the experts interviewed, there was a high level of self-consciousness concerning the role of Ghana in the international fight against AMR: 

“Ghana plays a leading role in the fight against AMR in the sub-region and also globally” (R4). “But …I think countries like Kenya, South Africa, okay, and perhaps Ethiopia are overtaking Ghana, right when it comes to implementation… So, we need to step up”.(R1)

“The Call to Action Conference that was funded by Wellcome and the other partners, was really really great. I mean, it brought in a lot more stakeholders, you know, experts who are working in that area, to share ideas…. Usually at the conferences of the International Pharmaceutical Federation, students who have done [work] with the AMR space have had the opportunity to showcase their work”.(R1)

There were, however, more voices insisting that Ghana should not repose on the international acknowledgement received in the past. There were concerns that the country was losing ground.

“So far, research and conferences are more geared towards trying to achieve targets for the development of researchers. You are under pressure to publish or perish. So, regarding policy implementation and informing policy that is on our mind, we try to do that, but we focus more on publish or perish. So much more has to be done regarding impact on policy”.(R5)

“… if you look at GLASS reporting… Ghana has not reported anything yet… I think we have worked very hard to be putting the systems and infrastructure in place and probably when we take off we would have smooth systems running. So, we contribute, attending international conferences for most of the international engagement; on the African continent and beyond. For me, I mostly have the opportunity to be sharing data… but that is not enough as far as I am concerned. We must hit the ground and get the evidence. The evidence at the moment is not there, so it is like we are not doing anything”.(R11)

The international role of Ghana has been based on close cooperation with multilateral organisations:

[Role of WHO, FAO, OIE?] “I think without them we would have just been sitting without doing anything. They have been very pressing and pushy. These abilities have gotten us this far. We alone would have been standing aloof”.(R5)

#### 3.2.10. Progress Reports and Evaluation of the NAP (2017–2021): Preparations for a NAP 2?

Considering the assessment by most respondents that in spite of the important preparatory work on the NAP, cooperation between institutions facilitated through the NPAR and the significant output of biomedical research, Ghana has “not made as much progress as some other countries” (R1), one should expect a growing interest in evaluating the progress and the obstacles during the implementation of the NAP. There has been a mid-term rapid review of the implementation of the NAP, which was not seen as a profound accomplishment by the respondents who referred to it: 

[Enforcement of existing regulations?] “… during the mid-term review, the AMR secretariat did a rapid review on the implementation of the NAP, and one of the main activities is enforcement of regulation on antimicrobial use in Ghana. No work has started on that arm”.(R2)

“In fact, I happen to be part of a team that did an evaluation of the NAP. We did it early last year. What we did was that we looked at the various actions and evaluated the progress made. All focal persons from the various institutions were on board. So, you ask yourself, this one have we done it, this one have we not done, why? Then we raise the questions; if we have done it, to what level, what is left? It was teamwork, I will say that the focal points of the various institutions involved did their own evaluation. The thinking was that an external person will be contracted to do the evaluation again, but you know all these things require financial resources to undertake so I don’t think that has happened”.(R10)

The AMR Secretariat, with the support of WHO, conducted a mid-term rapid review of the implementation as recommended by the NAP, but with limited methodological effort as described by R10). Results were published by the WHO Ghana Office [51], but not in any internet document published by Ghanaian government sources. The report showed 34% of the activities were on-going, 60% had no funding and therefore no attention, and only 6% of the activities were completed. This situation explained, among others, the problems concerning the delivery of comprehensive data (such as missing country-wide surveillance data; no data for the GLASS report). The status of each of the subjects is presented in Figure 3. The review recommends the commissioning of an independent external reviewer to conduct an in-depth assessment. However, there is no evidence that this has been carried out (see a remark quoted above from R10). The WHO report stressed the support of WHO to strengthen governance on AMR activities; IT equipment was presented to the AMR Secretariat, as well as technical support “to develop terms of reference for the Secretariat and the Interministerial Committee to strengthen governance mechanisms.”

Few respondents reflected on the fact that the NAP expires in 2021 and that there is the need for a new plan for the following years: 

“…because the plan that we have currently is expiring and we need a new plan. The problem is, you know, they will be doing some monitoring and evaluation, it will be highly technocratic, they will hire some consultation, they’ll do the evaluation but you’ll be very surprised to learn that real consultation in Ghana is often conducted in a manner that, to me, suggest it is perfunctory; they are just checking the box. And I think a lot of that problem persist. I don’t blame the technocrats at specific ministries or bureaucracies alone. I think it is just a general culture that we have in the country that makes accountability just difficult”.(R12)

“I can say our implementation is quite slow from where I sit. If you look at it critically people are prepared to work but everything is about money and if government should put in some money, because I can say the government has taken some of the NAP into the NDPC Plan. And you know, the president also co-chairs the SDGs, …, I remember … he asked that some of the indicators be pushed into SDGs and the National Development... forty-year plan” (R9; reference to Long-term National Development Plan of Ghana (2018–2057), of which only an Outline is available online [52].

Hardly any links to a future policy on AMR can be found online. On the webpage of the NDPC (National Development Planning Commission) there is an entry on Social development/ health and health services with no further links, but in “Medium-term Plans–Sectors” there is no entry pertaining to more recent developments, other than “Health Sector Medium-Term Development Plan 2014–2017”. There is a Medium Term Expenditure Framework (MTEF) for 2018–2021 from the Ministry of Health without any mention of AMR measures [53].

In a recent text by the MoH, “National Health Policy” (revised on January 2020) [54], “The Policy shall collectively ensure that there will be improved alignment, complementarity and synergies within and across all public sector ministries, as well as other stakeholders towards the achievement of the national health goal” (ibid., p. 7), only one small section refers once to antimicrobial resistance (under the heading “3.1.4 Strategy: Ensure the availability and appropriate use of quality medicines and medical products”), saying: “This will be done through strengthened regulation and the promotion of local production of these medicines and medical products as well as working within the One Health strategic framework to combat antimicrobial resistance” (p. 21). In his foreword, the Ghanaian president refers to the “Coordinated Programme of Economic and Social Development Policies (2017–2024)” [55] and a “Global Action Plan for Healthy Lives and Well Being” (related to SDG3, initiated by Ghana, Germany and Norway and committing the most important international organisations in this field to stronger cooperation) [56], but these documents also do not contain links to AMR policies.

Considering the work on other internationally demanded national action plans, such as the National Action Plan for Health Security (NAPHS) launched in 2020 [57], and work on the Global Health Preparedness Programme (GHPP) linked to the International Health Regulations and closely related to the issue of infection control [58], could it be that obligations by international organisations are creating a kind of overload for comparatively poorer states, which makes it difficult to implement action plans responding to these obligations?

## 4. Discussion

Ghana’s NAP on AMR has been guided by WHO standards (Global Action Plan on Antimicrobial Resistance and other WHO documents). Though many aspects of it have proven to be challenging to fully implement in the short term, it has constituted an important basis for a continued discourse among stakeholders within a broad institutional framework. To a large extent, this framework is supported by the OH concept, linking the whole stakeholder network in the field of AMR and creating a broad awareness among professional stakeholders about strategies to control AMR.

Antimicrobial stewardship has been acknowledged as key to combatting AMR. In Ghana, the regulations, policy documents and guidelines have been well established. However, our findings show that due to the poor coordination, reporting and publication among the various stakeholders, the progress made on the implementation of the NAP is fragmented. Moreover, based on the international accolade received during the development and launch of the Ghana NAP, the fact that Ghana had not yet submitted any report to the WHO Global Antimicrobial Resistance Surveillance System (GLASS) leaves much to be desired relative to other African countries such as Tanzania which also launched their NAP in the same year as Ghana [17]. Fighting the menace of AMR requires strong political commitment and relevant governance mechanisms at the national and regional level [59]. In a stakeholder analysis of the implementation of the NAP in Ghana, Jimah and Ogunseitan [60] found a strong political will to promote multi-sectoral partnership towards efforts to implement the NAP. This political will, however, needs to be translated into tangible actions. Although, some collaborations existed, the importance of using the OH approach allowed the involvement of more actors for more effective collaborative actions in the implementation of the NAP. Findings from the expert interviews in this study corroborated the expectation that the extent of coordination activities and of awareness formation among professionals has in fact been underestimated due to a lack of publicly available documentation. Cooperative work has played a larger role than is visible online, and small investments in improving the transparency could already have a larger impact of AMR policies on national politics in general.

The results from the interviews and literature on recent developments point to the following key findings, which demonstrate that both accomplishments and shortcomings regarding the Ghana NAP are ambiguous. They point to insufficient financial resources, stakeholder coordination, infrastructure and human capacity, awareness creation and the impact of a lack of communication on AMR policies.

Practical outcomes of the implementation process for the NAP have been limited due to (1) the lack effective enforcement of regulations, (2) limited implementation of larger projects because of a lack of financial resources provided by the Ghanaian government and (3) insufficient supplementation of efforts by international cooperation. These challenges have been highlighted in Ghana and in other developing countries [6,17,18,60,61]. These pitfalls reveal the need for developing countries to be mindful in the adoption of international policies and programmes. Using the WHO’s global action plan on AMR as a blueprint for the design of national action plans for AMR containment, developing countries should tailor the objectives and actions to local contexts and show strong political commitment beyond national discourse. In addressing the challenges identified, governments need to actively involve all stakeholders, including the general public, and strengthen collaboration at all levels of the NAP governance structure. Detailed monitoring and evaluation plans focussing on cost-effectiveness of specific policy interventions should be carefully planned and coordinated, and the results made public to instigate confidence in the population and external partners. Findings from this study scarcely reveal any “grey literature” such as position papers by the stakeholders and reports on meetings of the central institutions (National Platform on AMR; AMR Secretariat). Most importantly, the allocation of local resources with a transparent political process and accountability will enhance commitment and promote the sustainability of the NAP interventions.

For many countries in sub-Saharan Africa, there is a lack of adequate infrastructure and capacity for robust AMR surveillance systems [62]. The implementation of AMR NAP in developing countries depends on high-tech solutions that are user friendly at the primary level. In Ghana, these solutions have been initiated (mostly with international cooperation), but have not yet been comprehensively implemented, such as the nationwide compatible digitalisation of surveillance data on AMR prevalence and antimicrobial use and the full installation of the Logistics Management Information System (GhILMIS). With the successful roll out of mPedigree in detecting fake counterfeits and the diversion of products, including antimicrobials in other countries, Ghana could still leverage on this home-grown solution to complement the fight against the import of fake and substandard medications.

There is also a need to continuously improve the qualification of medical and other personnel involved in activities related to the NAP. To promote the effective implantation of the national strategies on AMR in Thailand for instance, technical capacities among implementing agencies were highlighted as essential to translate policies into practice [63]. However, considering high international inequalities, this also increases the risk of brain drain.

Concerning the critical analysis of the lack of explicit reference to AMR in many health-related documents, we found a certain caveat. Attention to AMR is indeed implicitly included in a number of other health-related programmes. This, however, reduces the visibility of AMR policies to the general public and also in international communication. It also reduces the transparency concerning the implementation of the NAP. Similar to other developing countries [17,18,63,64], the implementation of the AMR NAP has been fragmented and without a visible concerted effort between all stakeholders. The lack of a comprehensive multi-sectoral approach to the operationalisation of the NAP makes it difficult to paint a clear picture of the accomplishment of the action plan and an even more difficult task to attract both internal and external funding support. The sustainability of interventions, whether already initiated or planned, will remain questionable if efforts are not made to make outcomes more visible.

Awareness has considerably grown in professional groups beyond the original field of human health, which now particularly includes the agricultural sector, animal health and the environment. However, awareness remains weak in the rest of the population [60]. Our findings show that the current approaches used in sensitizing the public on the risk and challenges of AMR have not been effective in reaching the majority of the population, hence limiting the desired impact. Mainstreaming a comparatively complex topic as a focus of social attention should be seen as a long-term process, which has to be supported by consistently linking this to problems which affect people’s everyday life. Significant efforts are needed for raising and maintaining public awareness and interest, such as more direct community engagements during the WAAW observation and beyond as well as periodic sensitisation of the general public using all available media channels including peer-to-peer,. At the national and institutional levels, establishing AMR curriculums in our educational institutions and creating recurring opportunities for continuing education programmes for various professions and civil society organisations will provide an avenue for constant AMR/AMU education to new generations, but this again is a matter of scarce financial means.

The Ghanaian experience as well as the implementation problems of NAPs on AMR in other low- and middle-income countries point to the need for further research on their implementability regarding the time and the space dimensions. The formulation of global action plans by international institutions should consider the national preconditions for implementations, which differ significantly in the global context, and related to that, the specific time frames which are required to implement different elements of an action plan to fight AMR. Sequencing the implementation implies a step-by-step approach based on periodic strong evaluations of what has been achieved on specific goals, and it demands a longer-term perspective than just one five-year plan.

## 5. Conclusions

The lack of financial resources and a lack of full (and more explicit) integration of AMR policies into the Ghanaian health system and in Ghanaian society in general can be seen as the most important factors preventing a comprehensive implementation of all NAP dimensions. Improving diagnostics, institutional healthcare and better-organised access to medicines in marginal regions also would reduce the incentives to use informal markets for accessing (frequently sub-standard) antibiotics without prescriptions. The effective functioning of the National Health Insurance System (NHIS) (i.e., comprehensive health system, resources for high quality medical services, handling of data on surveillance through electronic systems) could improve the health-seeking behaviour of the general population and translate into better enforcement of existing regulations. This implies the task of mainstreaming the societal embedding of fighting AMR, but is also a general problem of the level of national per capita income and budgetary allocation to health interventions.

## Figures and Tables

**Figure 1 antibiotics-11-00613-f001:**
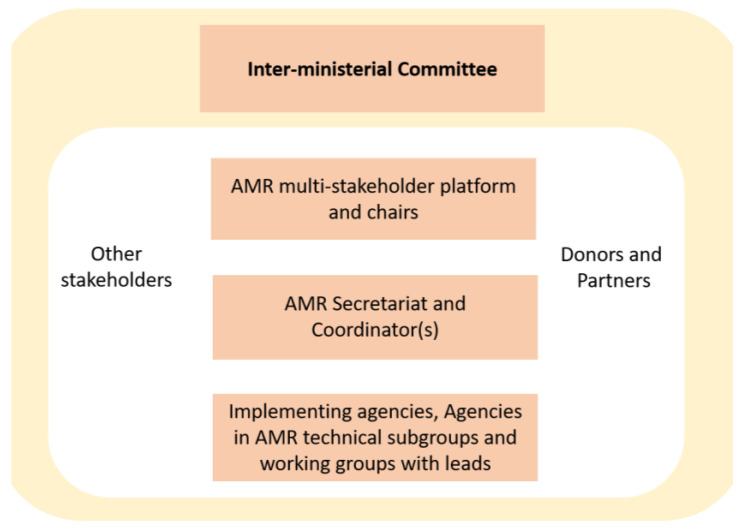
AMR Country Level Governance Framework. Source: Own design, based on Ghana National Action Plan, p. 7.

**Figure 2 antibiotics-11-00613-f002:**
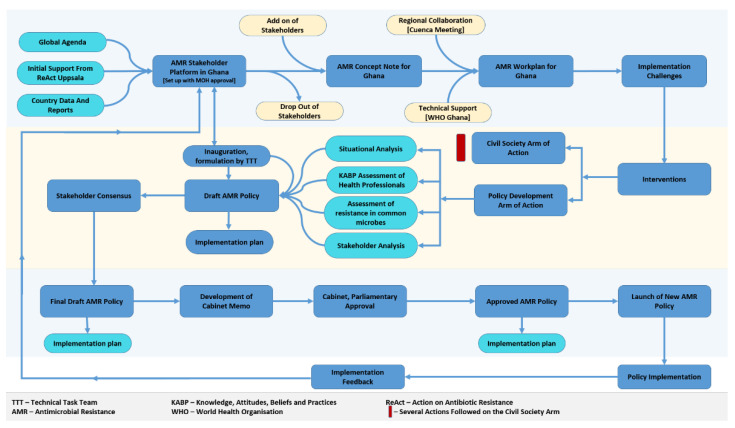
AMR Policy Flow. Own design, based on [15].

**Figure 3 antibiotics-11-00613-f003:**
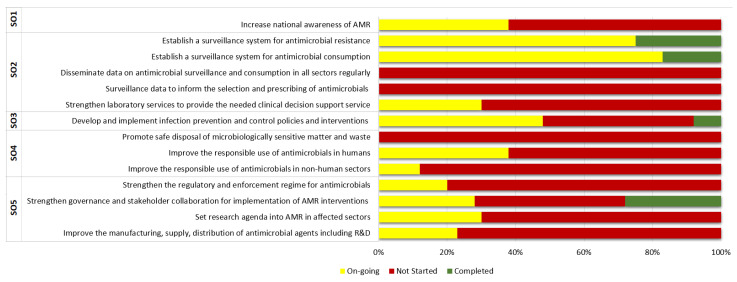
Level of implementation of Ghana NAP on AMR, Mid-term review 2020. Own design, based on [51].

**Table 1 antibiotics-11-00613-t001:** AMR Monitoring and Evaluation Framework. Source: Own design, based on Ghana’s National Action Plan, p. 59.

	Goal One	Goal Two	Goal Three	Goal Four	Goal Five
Strategic Objective	Improve awareness and understanding of antimicrobial resistance through effective communication, education and training.	Strengthen knowledge and evidence base through surveillance and research.	Reduce the incidence of infection through effective sanitation, hygiene and infection prevention measures and good agricultural and biosecurity practices.	Optimise the use of antimicrobial agents in humans, aquaculture, plant production and animal health in the “One Health” approach.	Develop the economic case and create an enabling environment for sustainable investment that takes into account Ghana’s needs, and increase investment in new machines, diagnostic tools, vaccines and other interventions.
Outcome Statement	Awareness and knowledge of AMR is improved.	Evidence-based knowledge to reduce the burden of AMR increased.	Occurrence of infections in establishments reduced.	Use of antimicrobials in animal and human health optimized.	Enhance the enabling environment for sustainable investment of AMR reduction.

**Table 2 antibiotics-11-00613-t002:** Affiliation of interviewed experts.

No. of Expert	Institution(s)	Sector(s)
R1	Kwame Nkrumah University of Science and Technology; National Platform on Antimicrobial Resistance	Research and academia
R2	AMR Secretariat; Ghana Health Service	Health
R3	Komfo Anokye Teaching Hospital; National Platform on Antimicrobial Resistance	Health
R4	Food Agriculture Organization	International agency
R5	Council for Scientific and Industrial Research, Water Research Institute	Research and academia
R6	Ministry of Health	Health
R7	Hope for Future Generations; member of various Civil Society Organisations (CSOs)	Non-governmental organisation
R8	Antibiotic Drug use, Monitoring and Evaluation of Resistance in Ghana (ADMER); Danish International Development Agency (DANIDA)	International agency
R9	World Health Organisation Country Office Ghana; Ministry of Health; Ghana National Drug Information and Resource Centre	International agency/ Health
R10	Environmental Protection Agency	Environment
R11	University of Ghana; National Platform on Antimicrobial Resistance; Antibiotic Drug use, Monitoring and Evaluation of Resistance in Ghana (ADMER)	Research and academia/ International agency
R12	IMANI Centre for Policy and Education	Non-governmental organisation
R13	Ministry of Food and Agriculture	Agriculture and veterinary

## Data Availability

In-depth interview guiding questions used to generate information for the study and interview minutes are available on request from the corresponding author.

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
