# Peer review of "Fighting Antimicrobial Resistance: Development and Implementation of the Ghanaian National Action Plan (2017–2021)"

_antibiotics, 2022, doi:10.3390/antibiotics11050613_

Round 1

Reviewer 1 Report

In this study, the authors (Fighting AntimicrobiaI Resistance: Development and Implementation of the Ghanaian NAP on AMR (2017-2021)). that AMR policies, as embodied in the NAP, have led to an extended network of cooperation between stakeholders in many political fields. Broadly, limited allocation of financial resources from the government and from international cooperation, have been deplored. Furthermore, the opportunity of using the NAP for mainstreaming the response to the threat of AMR has not been seized.

For the international readership, the topic is really interesting, as practical and theoretical implications are mentioned.

In the abstract I suggest to write that Ghana is one of the leading players, NAP although it is known but you need to clarify the abbreviation as national action plan.

In the introduction a more in-depth look in the recent regulation framework is recommended.

I was hoping to see a further prospective clinical study of the effects of the used could better support the claims.  

The results are too long while the discussion is too short, the figures could be better explained in the main text. I suggest add some measures of validity and reliability.

Author Response

Thanks for your comments.

(1) Abstract: We say "a leading player" (not "the") - so that's the same as "one of the leading...". We added "national action plan".

(2) We added some information on the regulatory framework.

(3) We refer at different places to clinical studies on Antimicrobial consumption and antimicrobial use. and to the lack of comprehensive studies at the national level. Problems which have so far prevented the realization of such studies are referred to. A more concrete proposal for designing such research would be beyond the scope of this paper.

(4) We condensed the first section of "Results", moved some interpretative sentences to "Discussions" and explained our statements in the discussion section. We clarified the background of Figure 3. We did not really understand your suggestion to "add some measures of validity and reliabilty" taking into account the character of our method.

Reviewer 2 Report

The paper by Wolfgang Hein et al., titled "Fighting AntimicrobiaI Resistance: Development and Implementation of the Ghanaian NAP on AMR (2017-2021)", discusses how Ghana has faired in the recent implementation of AMR policies, including the National Action Plan on antimicrobial Resistance (NAP AMR)

In their academic discourse, they report that there are policies on AMR, albeit with limited funding from Government and International stakeholders.

They make a call to integrate NAP AMR in other health and environmental programs to achieve more impact.

The manuscript can be improved by making the following recommendations

Generally, the intention of the paper is well written
But the detailed text can reduce the interest and readability of the manuscript. Consider re-organizing text between sections, moving issues to the appendix and to improve readability

1. In the title, avoid abbreviations even if they are common words. Also, antimicrobial Resistance and AMR are similar words phrases that appear twice in the title.
Please recast your title to avoid repetitions and abbreviations.

2. The reference citation style may need to be reviewed in line with the journal specifications,

3. When Citing articles, it may not be necessary to mention specific page numbers. 
4. In the introduction, there are a few abbreviations that need to be spelled out on first use example, NAP, ADMAR, REACT, NGO
5. Table 1 Caption write M&E in full
6. Line 127 instead of the Word the following Figure just cite Figure #
7. Figure 3: Add a full caption and also define the abbreviation in the figure

8. Line 180 -182, no reason to argue why a particular method was not used. This can be stated as a study limitation instead.
9. Under emerging themes, lines 235-326. Focus on discussing the arising themes and avoid methodological description to reduce the amount of text

10 Line 280 - 285, you start a discussion with certain authors. This can go into discussion. Your results need to focus on the findings/themes that cane out of your research
11: Table 3 can be avoided or sent to supplementary material. A table can not be pasted as a figure/picture
12 Line 333 to 352 agaib=n narrow down to themes found and avoid cross-referencing documents until in the discussion section
13 Same concern up to lines 414 [please separate methods and discussion text in this results section.
14. Figure 3: The results from figure 3 need to be explained as to how they were measured in the methodology
15. Again, up to line 932, I see discussions that do not belong to the RESULTS section and need to be moved out into the discussion

16. In the discussion, reduce the numbered or bulleted statement into more concise narratives- all this would improve readability to the consumer of this paper
17. The conclusion can also be shorted, and avoid quoting figures or references in this section.

Author Response

Thank you for the review. We made the following changes:

(1) We modified the title correspondingly.

(2) We tried our best to stick to the Antibiotics  citation rules.

(3) We omitted the page number in some cases. When there is an exact quote, we think, the page number should be included.

(4) We included the full names in a few cases, where they were not spelled out.

(5) Okay

(6) Okay

(7) Okay

(8) Slight change in the text following your proposal.

(9) We substantially shortened this section according to your proposal.

(10) These line are omitted from the main text.

(11) Table (3) now in supplementary material

(12) Line 333ff.: First paragraph omitted; text on Fleming Fund Grant included because this is central (in quantity as well as objectives) for international contributions.

(13) Same as (9)

(14) Figure (3): "Method" is explained and implicitly criticized by Respondent (10) a few lines above the figure; these, however, are the only (close to) quantitave estimates of the "level of implementation" which we could find in the net.

(15) In some cases we followed your advice to move elements of discussion from the results to the discussion section. As we tried to merge information from the interviews with most recent net-based information, in some cases it is necessary to include some linking statements to establish "the state of affairs" related to the issue discussed.

(16) We followed your proposal.

(17) Well, the conclusion was already short, but we followed your proposal to leave out the last sentence with the figures.
